

# A vertical representation of soil carbon in the JULES land surface scheme with a focus on permafrost regions

Eleanor J. Burke[1], Sarah E. Chadburn[2,3], and Altug Ekici[2,4]

[1]Met Office Hadley Centre, FitzRoy Road, Exeter, EX1 3PB, UK
[2]University of Exeter, College of Engineering, Mathematics and Physical Sciences, Exeter, EX4 4QF, UK
[3]University of Leeds, School of Earth and Environment, Leeds, LS2 9JT, UK
[4]Uni Research Climate and Bjerknes Centre for Climate Research, Bergen, Norway

*Correspondence to:* Eleanor Burke (eleanor.burke@metoffice.gov.uk)

**Abstract.** An improved representation of the carbon cycle in permafrost regions will enable more realistic projections of the future climate-carbon system. Currently JULES (the Joint UK Land Environment Simulator) - the land surface model of the UK Earth System Model (UKESM) - uses the standard 4-pool RothC soil carbon model. This paper describes a new version of JULES (vn4.3_permafrost) in which the soil vertical dimension is added to the soil carbon model, with a set of four pools

in every soil layer. The respiration rate in each soil layer depends on the temperature and moisture conditions in that layer. Cryoturbation / bioturbation processes, which transfer soil carbon between layers, are represented by diffusive mixing. The litter inputs and the soil respiration are both parameterised to decrease with increasing depth. The model now includes a tracer so that selected soil carbon can be labelled and tracked through a simulation. Simulations show an improvement in the large-scale horizontal and vertical distribution of soil carbon over the standard version of JULES (vn4.3). Like the standard version

of JULES, the vertically discretised model is still unable to simulate enough soil carbon in the tundra regions. This is in part because JULES underestimates the plant productivity over the tundra, but also because not all of the processes relevant for the accumulation of permafrost carbon are included in the model. In comparison with the standard model, the vertically discretised model shows a delay in the onset of soil respiration in the spring, resulting in an increased net uptake of carbon during this time. In order to provide a more suitable representation of permafrost carbon for quantifying the permafrost carbon feedback

within UKESM, the deep soil carbon in the permafrost region (below 1 m) was initialised using the observed soil carbon. There is now a slight drift in the soil carbon ($< 0.018$ % decade$^{-1}$), but the change in simulated soil carbon over the 20$^{\text{th}}$ century, when there is little climate change, is comparable to the original vertically discretised model and significantly larger than the drift.



## 1 Introduction

Soils contain the largest terrestrial carbon store, estimated at around 2000 Pg in the top 2m of soil (Batjes, 2016; Shangguan et al., 2014). In particular, permafrost regions contain a large amount of soil carbon, much of which is old carbon that is prevented from decomposing due to the frozen conditions (Schirrmeister et al., 2002; Zimov et al., 2006; Smith et al., 2004).

The most recent estimate suggests that there is approximately 1035 Pg carbon in the top 3 m of permafrost soil, and another 272 Pg carbon below 3 m in e.g. yedoma deposits (Hugelius et al., 2014). This relatively inert carbon has a critical role to play in the terrestrial feedbacks to climate change, as it decomposes when permafrost thaws, releasing greenhouse gases to the atmosphere and amplifying climate warming (Schaefer et al., 2014; Schuur et al., 2015; MacDougall et al., 2012; Burke et al., 2012, 2013; Schneider von Deimling et al., 2012, 2015). Current estimates suggest that there will be 35 - 205 Pg of permafrost carbon

emissions by 2100 (Schuur et al., 2015; Schaefer et al., 2014). However, the magnitude and timing of carbon fluxes caused by permafrost degradation remain highly uncertain, partly because of incomplete observations and partly because modelling of many of the relevant processes is still in its infancy.

Earth System Models (ESMs) play an important role in understanding feedbacks and global impacts - aiming to include all significant links between different components of the Earth system. There is currently a considerable uncertainty in the soil

carbon cycle feedbacks, and size and response of soil carbon pools to a changing climate in ESM simulations. For example in the CMIP5 models the soil carbon stocks correlate poorly with observations (Todd-Brown et al., 2013; Anav et al., 2013), which is very likely due to missing processes in the models (Todd-Brown et al., 2013). This in turn impacts the sensitivity of soil organic matter to environmental change, leading, for example, to the wide range of estimates of permafrost carbon emissions under future warming (Schuur et al., 2015). Only a few studies have explicitly included permafrost carbon coupled

with the climate in an ESM, e.g. MacDougall et al. (2012); MacDougall and Knutti (2016).

Recent developments in permafrost physics such as including soil freezing, organic soil properties, improved snow schemes, more realistic soil depths and physical impacts of mosses and lichens (Gouttevin et al., 2012; Lawrence et al., 2008; Ekici et al., 2014; Paquin and Sushama, 2014; Chadburn et al., 2015a, b; Porada et al., 2016) mean that the rate of permafrost thaw is now more realistic in many of the land surface components of ESM's. Adding a vertical representation of soil carbon is

now required to enable a representation of permafrost carbon in ESMs (Tian et al., 2015). Without a vertical representation, decomposition rates are determined only by soil temperatures above the maximum summer thaw depth, so the very slow turnover of deep carbon in the permanently frozen soil is not represented. Vertically resolved soil carbon and nitrogen have recently been introduced into the land surface schemes from several ESMs (Koven et al., 2013, 2009; Jafarov and Schaefer, 2016; MacDougall et al., 2012), some of which will participate in CMIP6. Other vertically resolved soil carbon models have

been applied on a site scale (Herbst et al., 2008; Braakhekke et al., 2011), with a view to being included in ESMs in future. It should be noted that any model that is included in an ESM must be applicable globally as well as for permafrost regions.

Typically soil carbon within an ESM is "spun-up" using pre-industrial climate until the soil carbon is relatively stable between spin-up iterations. However, models are often missing many relevant burial processes such as alluvial sedimentation; dust deposition; peat development and cryoturbation (Schuur et al., 2008; Dutta et al., 2006; Ping et al., 2015). Therefore





there will be biases in the soil carbon which will impact projections of permafrost carbon emissions (Foereid et al., 2012). One method of reducing these biases is to initialise the soil organic carbon stocks using the observed soil carbon distribution (Schneider von Deimling et al., 2015; Jafarov and Schaefer, 2016). However, this may result in a significant drift back towards the equilibrium state, and thus it is important to check that this drift is not so large as to mask the climate signal.

The purpose of this paper is to describe and evaluate a new vertically resolved soil carbon scheme integrated within the Joint UK Land-Environment Simulator (JULES at vn4.3_permafrost), which is the land-surface component of the UK Earth System Model (UKESM). We describe the model structure and evaluate the results of global simulations over the 20[th] century against observations of soil carbon stocks and respiration fluxes. The results are also compared with the original standard zero-layer soil carbon scheme. Although the ability of the vertically discretised soil carbon model to represent the distribution of soil

carbon is globally relevant, the current assessment focuses particularly on permafrost regions.

## 2    Materials and Methods

### 2.1    JULES model description

JULES is the land surface component of the new community Earth System model, UKESM (Jones and Sellar, 2015). It can also be run offline forced by observed meteorology at a regional or point scale as well as globally. JULES is described in Best et al.

(2011) and Clark et al. (2011). It is a community model with many users and ongoing developments: For recent developments see e.g. Harper et al. (2016); Chadburn et al. (2015a).

JULES includes a dynamic vegetation model (TRIFFID), surface energy balance, a dynamic snowpack model (one dimensional), vertical heat and water fluxes, soil freezing, large scale hydrology, and carbon fluxes and storage in both vegetation and soil. It also includes specific representations of crops, urban heat and water dynamics, fire diagnostics and river routing.

### 2.2    Model Developments

#### 2.2.1    RothC soil carbon model

The standard soil carbon model in JULES is based on RothC (Jenkinson et al., 1990; Jenkinson and Coleman, 1999), and described in detail in Clark et al. (2011). There are 4 pools: decomposable plant material (DPM), resistant plant material (RPM), microbial biomass (BIO) and hummus (HUM). The soil carbon dynamics are represented as follows:

$$\frac{dC_{DPM}}{dt} = f_{dpm}\Lambda_c - R_{DPM} \tag{1}$$

$$\frac{dC_{RPM}}{dt} = (1 - f_{dpm})\Lambda_c - R_{RPM} \tag{2}$$

$$\frac{dC_{BIO}}{dt} = 0.46\beta_R R_{tot} - R_{BIO} \tag{3}$$





$$\frac{dC_{HUM}}{dt} = 0.54\beta_R R_{tot} - R_{HUM} \tag{4}$$

where $R_{tot} = R_{DPM} + R_{RPM} + R_{BIO} + R_{HUM}$ is the total respiration in kg C m$^{-2}$ s$^{-1}$, $t$ is the time in s, the $C_i$ are the carbon pools in kg C m$^{-2}$, $f_{dpm}$ is the fraction of litter that goes into DPM (dependent on vegetation type), $\Lambda_c$ is the total litter input in kg C m$^{-2}$ s$^{-1}$ and $\beta_R$ is the fraction of soil respiration which is emitted to the atmosphere. This depends on soil texture.

The respiration for each pool ($R_i$, where $i$ is one of ($DPM$, $RPM$, $BIO$, $HUM$)) is given by:

$$R_i = k_i C_i F_T(T_{soil}) F_s(s) F_v(v) \tag{5}$$

where the $k_i$ are fixed constants in s$^{-1}$ (Clark et al., 2011). The functions of temperature [$F_T(T_{soil})$] and moisture [$F_s(s)$] depend on the temperature and moisture content near the surface. The function for vegetation $F_v(v)$ is a function of the vegetation cover. The moisture function in the current version of JULES (vn4.3) is parametrised as:

$$F_s(s) = \left\{ \begin{array}{lcl} 1 - 0.8(s - s_0) & ; & s > s_0 \\ 0.2 + 0.8\left(\frac{s - s_{min}}{s_0 - s_{min}}\right) & ; & s_{min} < s \le s_0 \\ 0.2 & ; & s \le s_{min} \end{array} \right\}, \tag{6}$$

where $s$ and $s_0$ are the unfrozen soil moisture content and the optimum soil moisture, both expressed as a fraction of saturation. $s_0 = 0.5(1 + s_w)$, and $s_{min} = 1.7 s_w$ where $s_w$ is the soil moisture at wilting point also as a fraction of saturation. There are two different functions available to represent the impact of temperature on soil respiration. The first option for the temperature function, $F_{T,Q10}$ (Equation 8), is a commonly used exponential function, and the second option, $F_{T,Roth}$ (Equation 7), is based on the function from the original RothC model.

$$F_{T,Roth}(T_{soil}) = \frac{47.9}{1 + e^{106/(T_{soil} - 254.85)}} \tag{7}$$

$$F_{T,Q10}(T_{soil}) = Q_{10}^{(T_{soil} - 298.15)/10} \tag{8}$$

Where $T_{soil}$ is the soil temperature in K and Q$_{10} = 2$. Figure 1 shows $F_{T,Q10}$ allows much more respiration at temperatures below freezing than $F_{T,Roth}$.

$F_v$ is a function of the vegetation fraction, $v$:

$$F_v(v) = 0.6 + 0.4(1 - v) \tag{9}$$

All of these modifying functions are poorly constrained, and of these the temperature function has the largest impact on the simulation (Bauer et al., 2008; Exbrayat et al., 2013). Therefore both versions of $F_T$ are evaluated in our simulations.





#### 2.2.2 Vertical discretisation

In the new, vertically discretised version of the soil carbon model there is a set of the four soil carbon pools ($DPM$, $RPM$, $BIO$, $HUM$) in every soil layer. The respiration rate is determined for each soil layer depending on the temperature and moisture conditions in that layer. Following Koven et al. (2013) we also add a vertical mixing (diffusion) term, with diffusivity $D(z)$ in m$^2$ s$^{-1}$ ($z$ is the vertical dimension in m). The equations for each soil carbon pool become:

$$\frac{\partial C_{DPM}(z)}{\partial t} = \frac{\partial}{\partial z}\left(D(z)\frac{\partial C_{DPM}(z)}{\partial z}\right) + f_{dpm}\Lambda_c(z) - R_{DPM}(z) \tag{10}$$

$$\frac{\partial C_{RPM}(z)}{\partial t} = \frac{\partial}{\partial z}\left(D(z)\frac{\partial C_{RPM}(z)}{\partial z}\right) + (1 - f_{dpm})\Lambda_c(z) - R_{RPM}(z) \tag{11}$$

$$\frac{\partial C_{BIO}(z)}{\partial t} = \frac{\partial}{\partial z}\left(D(z)\frac{\partial C_{BIO}(z)}{\partial z}\right) + 0.46\beta_R R_{tot}(z) - R_{BIO}(z) \tag{12}$$

$$\frac{\partial C_{HUM}(z)}{\partial t} = \frac{\partial}{\partial z}\left(D(z)\frac{\partial C_{HUM}(z)}{\partial z}\right) + 0.54\beta_R R_{tot}(z) - R_{HUM}(z) \tag{13}$$

The litter inputs, $\Lambda_c(z)$, now vary with depth. In reality, most of the litter enters at the top of the soil, but there is a smaller amount of litter input into deeper soil layers, for example from roots. In JULES, following Koven et al. (2013), we distribute the litter inputs declining exponentially with depth.

We modified the respiration terms in the new model version (from the original, Eq. 5), by including an extra decay of respiration with depth, based on Jenkinson and Coleman (2008) and Koven et al. (2013). This accounts for factors that are currently missing in the model such as priming effects, anoxia, soil mineral surface and aggregate stabilization. The respiration terms are now:

$$R_i(z) = k_i C_i(z) F_T(T_{soil}(z)) F_s(s(z)) F_v(v) \exp(-\tau_{resp} z) \tag{14}$$

$\tau_{resp}$ is an empirical parameter (in m$^{-1}$) with considerable uncertainty. The larger the value of $\tau_{resp}$, the more inhibited the respiration is with depth. In an equilibrium version of the vertically discretised soil carbon model, it was shown that the soil carbon vertical distribution and total amount is strongly dependent on the value of $\tau_{resp}$, more so than any other model parameter.

The vertical mixing term in Equations $10-13$ represents either bioturbation - mixing of the soil due to animals and plant roots - or cryoturbation - where soil mixing occurs due to frost heave and freeze-thaw processes. The diffusion rate, $D(z)$, varies between grid cells and with depth. We follow Koven et al. (2013) by changing the rate depending on the presence of permafrost. Without permafrost, there is a constant bioturbation mixing rate of 1 cm$^2$ year$^{-1}$. When permafrost is present, the mixing represents cryoturbation and the rate increases to 5 cm$^2$ year$^{-1}$, but drops off linearly below one meter and reaches





zero at 3 m depth. Permafrost is diagnosed wherever the deepest soil layer is below 0°C, assuming that this layer is below the depth of zero annual amplitude.

Further modifications to these rates could be considered in future work. For example, bioturbation rates may vary with depth (Johnson et al., 2014). There are few explicit measurements of cryoturbation rates, but the available observations suggest that 5 cm$^2$ year$^{-1}$ may be a realistic value (Klaminder et al., 2014). However, further studies are required to better constrain soil mixing processes. Some modelling studies have incorporated depth-dependent bioturbation mixing, e.g. Vanwalleghem et al. (2013), and there are a few detailed models of cryoturbation, e.g. Peterson et al. (2003).

### 2.2.3 Adding a soil carbon tracer

In order to more explicitly study soil carbon dynamics in transient simulations, we added a tracer to the model. This works by labelling some of the carbon at the start of the simulation and keeping track of the labels at this carbon moves through the system, whether mixing into different layers or leaving the soil through respiration.

Each soil carbon pool in each soil layer is assigned a fraction, $Fr_{oldC}$, at the start of the main run, representing the fraction of carbon in that pool that is 'labelled'. This fraction is then updated whenever the soil carbon pools are updated, either due to input of fresh carbon from litter (which reduces the fraction), or due to mixing of carbon between two layers in which the fractions are different. The general formula to update the old carbon fraction ($Fr_{oldC}$) for carbon pool $C_i$ (kg m$^{-2}$), with an increment of carbon $C_i \rightarrow C_i + \Delta C_i$ is:

$$Fr_{oldC|C_i} \rightarrow \frac{Fr_{oldC|C_i}C_i + Fr_{oldC|\Delta C_i}\Delta C_i}{C_i + \Delta C_i} \tag{15}$$

$\Delta C_i$ includes both incoming and outgoing fluxes from the pool. For the outgoing fluxes in $\Delta C_i$, we assume that $Fr_{oldC}$ is the same as for the $C_i$ pool. For an incoming litter flux we assume that $Fr_{oldC}$ is zero, and incoming fluxes from other pools naturally take the $Fr_{oldC}$ value from the corresponding pool. The fraction of labelled carbon in the outgoing respiration flux is also output from the model.

Multiplying the carbon pools/fluxes by their corresponding fraction, $Fr_{oldC}$, gives the quantity of labelled carbon in the pool/flux, allowing the user to follow it through the system.

The choice of which carbon is labelled and traced through the system depends on the scientific question. For example, any carbon that is in permanently frozen soil may be given a value $Fr_{oldC} = 1$ at the beginning of the simulation, and carbon in other parts of the soil given a value $Fr_{oldC} = 0$, allowing us to explicitly trace the permafrost carbon. In this paper we label all carbon below 1.0 m with a value of 1 to study the behaviour of the deep soil carbon.

### 2.3 JULES simulations

Global simulations were carried out using a permafrost version of JULES 4.3 (JULES vn4.3_permafrost). This included the changes to the physical model described by Chadburn et al. (2015a, b). Developments include a representation of moss and organic soils and the addition of bedrock. In addition there was a higher resolution soil column with deeper soil. These modifi-





cations result is a reduction in the annual cycle of soil temperatures and a reduction in the summer thaw depth so that it better matches the observations over the standard configuration of JULES (vn4.3).

The simulations discussed here followed the protocol for the S3 experiments in TRENDY (Sitch et al., 2015). Forcing consisted of time-varying $CO_2$, climate from the CRU-NCEP data-set (v4, 1901-2012), and the fraction of agriculture in each grid cell (Hurtt et al., 2011). The model resolution was N96 (1.875° longitude x 1.25° latitude). Nine PFTs were used: tropical broadleaf evergreen trees (BET-Tr), temperate broadleaf evergreen trees (BET-Te), broadleaf deciduous trees (BDT), needleleaf evergreen trees (NET), needleleaf deciduous trees (NDT), C3 and C4 grass, evergreen shrubs (ESh), and deciduous shrubs (DSh). These were parameterised following Harper et al. (2016). Plant competition was allowed, with TRIFFID updating vegetation fractions on a daily time step.

Results using two alternative parameterisations of the soil respiration are shown here. The first one is denoted JULES-Q10 and uses $F_{T,Q10}$ (Equation 8), and $\tau_{resp} = 2$. The second one is denoted JULES-Roth and uses $F_{T,Roth}$ (Equation 7), and $\tau_{resp} = 1.2$. Respiration is more suppressed at depth in JULES-Q10 than in JULES-Roth leading to a greater proportion of soil carbon deeper in the soil profile in JULES-Q10 than in JULES-Roth. For comparison purposes, additional JULES simulations were carried out using the standard soil carbon model (vn4.3), which uses the temperature and soil moisture from the first layer of the soil to calculate one set of soil carbon pools representative of the whole soil profile. These standard simulations are denoted JULES-Q10$_{onelyr}$ and JULES-Roth$_{onelyr}$.

The soil carbon distribution is the slowest part of JULES to reach equilibrium. The 'modified accelerated decomposition' technique (modified-AD) described by (Koven et al., 2013) was used to spin it up to an initial distribution applicable for the year 1900. For the modified-AD the decay rates for the four pools were set to that of the fastest pool. In addition the diffusion coefficients for the different pools were multiplied by the same factors. An initial equilibrium spin up of 500 years was carried out to get the vegetation distribution and soil physical properties approximately correct. The model was then spun up by repeating the climate of 1901-1920 25 times. The decomposition rates and the diffusion coefficients were then reset, the soil carbon pools rescaled by the relevant factors and the model spun up until the change in soil carbon was less than 0.012 % decade$^{-1}$ globally and 0.005 % decade$^{-1}$ for the permafrost region.

## 2.4 Evaluation data sets

The Circum-Arctic map of permafrost and ground-ice conditions (Brown et al., 1998) gives a historical permafrost distribution, which can be compared with the permafrost area simulated by the model. It records continuous, discontinuous, sporadic, and isolated permafrost zones, for which the estimated permafrost coverage is 90-100%, 50-90%, 10-50% and <10% respectively. Since the model does not include sub-grid scale information, the simulated extent was compared with the continuous and discontinuous regions on the observed map.

There are no large-scale observations of litter available, but the annual total litter will be approximately the same as the annual total Net Primary Productivity (NPP). Observations of NPP are derived from MODIS data using the MOD17 algorithm (Zhao and Running, 2010). Here we assess the multiannual mean NPP for the period 2000 to 2012. Three notable biomes





were identified based on 14 World Wildlife Fund terrestrial regions (Olson et al., 2001; Harper et al., 2016): tundra; boreal and coniferous forest; and tropical forest.

There are two different large-scale observationally-based soil organic carbon datasets used for evaluation. The WISE30sec data set (Batjes, 2016) was created using the soil map unit delineations of the broad scale Harmonised World Soil Database,
version 1.21, with minor corrections, overlaid by a climate zones map as covariate, and soil property estimates derived from analyses of the ISRIC-WISE soil profile database (Batjes, 2009) for the respective mapped 'soil/climate' combinations. This is available for soil layer depths of 0-20 cm; 20-40 cm; 40-60 cm; 60-80 cm; 80-100 cm; 100-150 cm; and 150-200 cm. The Northern Circumpolar Soil Carbon Database Version 2 (NCSCDv2: Hugelius et al. (2014)) is more appropriate for the northern high latitudes because it includes more site observations than WISE30sec. It is, however, restricted to the northern
high latitudes and has a lower-resolution depth structure. NCSCDv2 consists of spatially extrapolated soil carbon data from more than 1700 soil core samples and gives soil organic carbon for the following depths: 0–30 cm, 0–100 cm, 100–200 cm, and 200–300 cm depth.

The multiannual mean and seasonal cycle of observed soil respiration for the period 2000 to 2012 was extracted from Hashimoto et al. (2015). Hashimoto et al. (2015) combined a global soil database with a semi-empirical model to scale up the
field observations of soil respiration to the global scale and provide a data-oriented estimate of soil respiration.

## 3    Results

The four different JULES simulations - the two standard simulations (JULES-Q10$_{onelyr}$ and JULES-Roth$_{onelyr}$ - vn4.3) and the two vertically discretised simulations (JULES-Q10 and JULES-Roth - vn4.3_permafrost) - all have the same soil physics and vegetation dynamics. The only differences between the simulations are in the soil carbon and soil respiration, which do not
feed back onto any of the other land surface processes when JULES is run 'offline' driven by observed meteorology (as here).

Figure 2 shows the JULES simulation of the mean permafrost extent for the period between 1961 and 1990, representative of the time period over which the observations were made. The simulated area is 20.3 million km$^2$ and the area of the discontinuous and continuous permafrost calculated in a similar manner from the Brown et al. (1998) data is 16.5 million km$^2$. This slight overestimation by JULES is caused by an overestimation in Eurasia, where the southernmost extent of the simulated
permafrost includes regions where there is only isolated or sporadic permafrost, which JULES is not expected to capture.

The addition of the vertical representation of soil carbon is most likely to impact model simulations in the permafrost region, because conditions there are very different in the deeper soil compared with the surface. However, the results must also be assessed both globally and in the tropics to ensure the model remains appropriate for use in a global Earth System Model (ESM). Results are presented for three regions: (1) global; (2) tropical (latitudes less than 23.5°); and (3) the region where
JULES simulates permafrost and NCSCDv2 has soil carbon data (outlined by a black contour in Figure 3).





## 3.1 Soil carbon stocks

The soil carbon quantity and distribution are highly dependent on the surface input of soil organic carbon, which comes from plant litter. Since there are limited observations of litterfall, the simulated Net Primary Productivity (NPP) was compared with observations (Figure 3). The large-scale spatial comparison between model and observations is visibly good with a spatial

correlation of 0.73. In much of the low and mid-to-high latitudes the simulated NPP is higher than that observed whereas in the drier and colder regions the NPP appears lower. These differences are summarised in Figure 4. Globally JULES overestimates the annual total NPP by $\sim$12 % compared with MODIS. Much of this overestimation occurs in the tropics. In particular the observed tropical forest biome (right hand bar plot) is about a third more productive in JULES than in the observations. In contrast, JULES underestimates the productivity of the permafrost region (highlighted by the black contours in Figure 3). In

general, NPP in the permafrost region is very low - the observed amount is 3.7 Pg C year$^{-1}$. The JULES-simulated value of 2.5 Pg C year$^{-1}$ is more than 30 % lower than observed. Most of this difference occurs in the observed tundra biome, where the simulated productivity is almost half that of the observations. As with the tropical forest, the boreal forest is slightly too productive, which for the permafrost region as a whole counteracts some of the simulated low-bias from the tundra. Errors in NPP will impact the simulated soil carbon distribution - too low NPP means too little input of organic carbon to the soil,

resulting in a low soil carbon and vice versa.

Table 1 shows the total soil carbon simulated by JULES for the different regions and biomes in the top 2 m of the soil. This can be compared with both the WISE30sec data set and the NCSCDv2 data set. In general both of the new vertically discretised JULES models perform better than the standard models when compared with observations. Their global total is more than twice that of the standard model versions and higher than the total in the WISE30sec data set. However, over the

NCSCDv2 region the WISE30sec data has 680 Pg C whilst NCSCDv2 has 1031 Pg C. Therefore WISE30sec simulates 351 Pg C less than NCSCDv2. Roughly combining these two data sets suggests that the global total could be nearer to 2300 Pg and therefore only slightly lower than the new model estimates. The biggest improvement in the vertically resolved model is the amount of soil carbon in the permafrost region. The standard JULES versions (JULES-Roth$_{onelyr}$ and JULES-Q10$_{onelyr}$) have far too little soil carbon in this region when compared with either the WISE30sec or the NCSCDv2 data. This increases

significantly, to a value comparable with the observations, when using JULES-Roth or JULES-Q10.

In terms of biomes (Table 1), all 4 versions of the model have a reasonable estimate of the soil carbon in the tropical forest biome. The vertical discretisation increases the soil carbon slightly. In the boreal forest biome the soil carbon is slightly lower than observations for the standard version of the model. This increases significantly for the vertically discretised models, leading to an overestimation of the soil carbon in the boreal forest by both model versions. The amount of soil carbon in the

tundra is significantly larger for the vertically resolved model versions and closer to both observational data sets, but remains too low. Some of the differences highlighted in Table 1 are caused by errors in the soil carbon input, i.e. NPP (Figure 4). However, particularly in the cold regions, the errors are also related to missing processes in the model such as dust deposition and peat accumulation, and by the fact that the soil carbon is not in equilibrium with the current climate - an assumption that was made when initialising the model.





The spatial distribution of the soil carbon in the top 2m of the soil is shown in Figure 5. On first glance the modelled spatial patterns are very similar, with higher levels of soil carbon in the boreal forest region, eastern America and Europe. Spatial correlations between the standard and layered soil carbon models are high (0.82 between JULES-Roth and JULES-Roth$_{onelyr}$ and 0.92 between JULES-Q10 and JULES-Q10$_{onelyr}$). However, spatial correlations between the model and WISE30sec
observations are lower at 0.40, 0.27, 0.30 and 0.22 for JULES-Roth, JULES-Roth$_{onelyr}$, JULES-Q10 and JULES-Q10$_{onelyr}$ respectively. There is a slight improvement in the spatial correlation with WISE30sec of the vertically resolved model compared with the standard model, but they remain relatively low. In comparison with the WISE30sec observations, the model has a greater area in the northern mid-latitudes with high values of soil carbon, and a greater area in the northern high latitudes and deserts with very small amounts of soil carbon. The vertically resolved simulations show more soil carbon in some of these cold
regions compared with the standard model, making it more comparable to the WISE30sec data than the standard simulations.

Figure 6 shows the zonal distribution of soil carbon for the permafrost region (highlighted in Figure 3) for both NCSCDv2 and WISE30sec data. The vertically discretised models are significantly improved over the standard model versions with much more soil carbon at latitudes where the observations show more soil carbon. There is still a mismatch between the latitude where the soil carbon is greatest - about 65° N for the observations, but only 60° N for the models. The low simulated soil
carbon in the region between 65 and 80° N is partly caused by the low simulated NPP in those regions (Figure 3).

Figure 7 shows the profile of soil carbon for the regions in this study. The WISE30sec data is only available down to 2m, whilst the NCSCDv2 is available from 0 to 3 m. In the permafrost region, the WISE30sec and NCSCDv2 have different profiles, with the NCSCDv2 having a greater proportion of its soil carbon between 1 and 2m than the WISE30sec data and a smaller proportion nearer the surface. In the modelled permafrost region (highlighted in Figure 2 in grey) the vertical mixing of
the organic carbon through the soil profile represents cryoturbation, whereas for the rest of the global land surface the mixing represents bioturbation, with a smaller mixing rate (see Section 2.2.2). The comparison of model and observations suggests that the model simulates too much soil carbon near the surface in the top 50 cm and not enough deeper in the soil both globally and in the tropics, which may be in part due to the representation of bioturbation. In the permafrost region, the model simulations approximately follow the shape of depth distribution of the WISE30sec data, although the soil carbon density is too low.

There are significant differences between the JULES-Roth and JULES-Q10 simulations. The $F_{T,Roth}$ rate-modifying function has a steeper gradient with temperature than the $F_{T,Q10}$ Figure (1), meaning that JULES-Roth tends to simulate less soil carbon in warm regions (very high decomposition) and more soil carbon in cold regions (very low decomposition), e.g. Table 1; Figure 5. These JULES-Roth results overall compare better with the observations, for example in Table 1, Figures 6 and 7. JULES-Q10 simulates too little soil carbon in the permafrost regions, and globally JULES-Roth has a better spatial correlation
with the WISE30sec data (0.4, compared with 0.3 for JULES-Q10).

### 3.2 Soil respiration

The addition of vertically discretised soil carbon has little impact on the magnitude of the soil respiration (Figure 8). This is expected because the climate is relatively stable at the beginning of the 21$^{st}$ century and the inputs (litterfall) are expected to approximately equal the outputs (respiration). Figure 8 shows that the seasonal cycle is very similar for all 4 model versions





in the tropics. However, in the permafrost region the seasonal cycle is slightly displaced, so that the peak of the respiration happens later in the year in the vertically resolved simulation. This shifts the peak so that it is approximately 20 days later for both JULES-Q10 and JULES-Roth. In JULES-Q10$_{onelyr}$ and JULES-Roth$_{onelyr}$, the respiration increases with the warming of the top soil layer in spring and reduces once the soil surface layer cools back down early in the autumn. In JULES-Q10 and JULES-Roth the respiration is dependent on the soil temperature profile - the soil warms up slowly from the surface downwards during late spring leading to a slower increase in respiration as the air temperature increases. At the end of the summer, the deeper soil layers cool more slowly than the surface, so respiration continues for longer. The delay in time of peak respiration is also notable in the total global soil respiration.

This change in the seasonal cycle of soil respiration impacts the seasonal cycle of Net Ecosystem Exchange (NEE - Figure 9). The annual amplitude of the global NEE increases in the vertically resolved models. They uptake more carbon in the northern hemisphere spring/summer and lose more in the northern hemisphere autumn/early winter. In the permafrost region the onset of carbon uptake is up to a month earlier in the vertically resolved model when compared with the standard model.

JULES-Roth and JULES-Q10 have a different seasonality. JULES-Roth has its peak uptake earlier in the year than JULES-Q10 and begins emitting carbon in August. This emission in August is because the soil respiration in JULES-Roth is higher than the NPP. JULES-Q10 has a smaller seasonal cycle of soil respiration and smaller maximum summer value, resulting in an uptake of carbon for a longer period during the northern hemisphere summer. The difference in the annual cycle of soil respiration between JULES-Roth and JULES-Q10 are due to the higher temperature sensitivity of the function $F_{T,Roth}$ compared with $F_{T,Q10}$. The larger seasonal cycle in JULES-Roth is closer to the observed amplitude on Figure 8.

The changes in the simulated global seasonal cycle of NEE, when included in the Earth System Model, will feedback onto the atmospheric $CO_2$ and impact any climate simulations.

### 3.3 Soil carbon changes over the 20$^{th}$ century

Figure 10 shows that the inclusion of vertically discretised soil carbon only has a small impact on the change in soil carbon over the 20$^{th}$ century. Globally, the simulated soil carbon increases by around 1 Pg C y$^{-1}$ for the period between 1960 and 2009, with some small differences between model versions. The increase is about 0.15 Pg C y$^{-1}$ greater for the vertically resolved model than the standard soil carbon model and is a consequence of a slight decrease in the overall sensitivity of soil respiration to temperature changes in the discretised model. The net global response is the combination of an increasing sensitivity of respiration to temperature changes in the warmer regions (tropics - Figure 10) and a decreasing sensitivity in the colder regions (permafrost - Figure 10).

In the permafrost region and between 1960 and 2009 the soil carbon increases by 146-168 Tg C. This falls within the modelled spread found by McGuire et al. (2016) who used a range of land surface models and showed the soil carbon increased over the permafrost region by 264 (42 − 637) Tg C y$^{-1}$ for the period 1960-2009. The slightly faster increase (by ∼10 to 25 Tg C y$^{-1}$) in the vertically discretised models are a consequence of the lower response of soil respiration to temperature change - possibly caused by a lag in the response of the deep soil temperatures to increasing air temperature. In the standard model, the respiration only responds to the surface soil temperatures, which will respond much more quickly to changes in air temperature





than the deeper soil. It should be noted that the difference in the soil carbon between the standard and vertically discretised model are small compared with differences between different models in e.g. McGuire et al. (2016).

The deep soil carbon was initialised in 1901 and tracked over the 20[th] century (Figure 11). Figure 11 shows the spatial distribution of the fraction of the total soil carbon that is labelled as "deep carbon", i.e. the ratio of the carbon below 1m depth

to the total soil carbon in the profile. In general JULES-Q10 has more deep soil carbon than JULES-Roth because it has more suppression of respiration with depth ($\tau_{resp}$=2 compared with $\tau_{resp}$=1.2). Both JULES simulations give a large proportion of deep soil carbon in the permafrost regions with 53 to 60 % at depths below 1 m (Table 2) and a much lower proportion over the rest of the land surface (particularly for the temperate and tropical regions where it is $\lesssim$ 20 %). These proportions are comparable with the proportion observed in the NCSCDv2 data set (59 %), although the absolute magnitude of the soil carbon

is too low. However, the observations are much more spatially variable than the model simulations (Table 2 and Figure 11), with more carbon in the deep soil in North America compared with Eurasia.

The change in the labelled deep soil carbon over the 20[th] century for the permafrost region highlighted is shown as a time series in Figure 11. Despite the increase in the soil carbon in the permafrost region (Figure 10), there is a decrease in the total labelled deep soil carbon in the soil profile of around 33-49 Tg C y$^{-1}$. A further 41-80 Tg C y$^{-1}$ labelled deep soil carbon

is mixed out of the deep soil and into the top 1 m of the soil. Vertical mixing processes in JULES continue to add new deep soil carbon resulting in a net increase in total deep soil carbon of $\sim$100 Tg C y$^{-1}$ over the 20[th] century. However, this deep soil carbon now consists of both 'original' permafrost carbon and 'newer' active carbon which, in reality, are likely to have different qualities (Harden et al., 2012).

### 3.4   Model initialisation of permafrost carbon

The new soil carbon model has improved the simulated soil carbon distribution for both versions of JULES. However, there are still considerable errors in the spatial distribution of soil carbon, reflected by the low spatial correlation between model and observations and most notable in the northern polar regions in (Figure 5 and Figure 6). This is caused, in part, by errors in the litter input to the soil - illustrated here as differences between the observed and modelled Net Primary Productivity (Figure 6). Errors in the litter input are more likely to cause errors in the soil carbon in the active layer which turns over at shorter

timescales. Any carbon frozen in the permafrost has been buried over several thousand years by alluvial sedimentation; dust deposition; peat development and cryoturbation (Zimov et al., 2006; Schuur et al., 2008; Ping et al., 2015). The only burial process included in the vertically resolved soil carbon model discussed here is cryoturbation. Therefore the model should not be expected to simulate the soil carbon stores introduced by these additional burial processes. However, errors caused by these missing processes will bias simulations of soil carbon and the response of the soil carbon to a changing climate. This will have

implications for any estimate of the permafrost carbon feedback when JULES is used within UKESM.

One method of incorporating a spatially realistic quantity of relatively inert permafrost carbon is to simply replace the simulated deep soil carbon below 1m with the observed soil carbon from the NCSCDv2 in the permafrost region ('PFC added: JULES-Roth' and 'PFC added: JULES-Q10'). JULES has four soil carbon pools, whereas the observations only give total soil carbon. Therefore the observed soil carbon was partitioned into the four pools based on the model's simulation of partitioning





for each grid cell. Table 2 shows the labelled deep soil carbon in the two JULES simulations is approximately equal to that in the NCSCDv2 data set. This has increased the total soil carbon in the permafrost region so it more closely represents the total observed soil carbon in the NCDSDv2. However, there is still a deficit of 122 to 283 Pg C depending on model configuration. The main differences are in the top 1m of soil, which is simulated by the model. Figure 12 shows the zonal total soil carbon for

5 the permafrost region. The thin red and black lines ('PFC added: JULES-Roth' and 'PFC added: JULES-Q10') show that with the addition of the observed soil carbon below 1m, JULES is closer to the observations, particularly in the region between 65 and 75° North. The global spatial correlation of the blended model and NCSCDv2 soil carbon with the WISE30sec (top 2 m) increases from 0.40 to 0.53 for JULES-RothC and from 0.3 to 0.43 for JULES-Q10.

 The addition of the permafrost carbon to the model could result in a significant drift back towards the equilibrium state. In

10 order to quantify the size of this drift the model was re-spun up from 1901-1920 for 25 x 20 years. After 500 years the global soil carbon had increased by 23 Pg C or 0.75 % of the global total for JULES-RothC and decreased by 2 Pg C or 0.06 % for JULES-Q10. The soil carbon in the permafrost region had increased by 7.5 Pg C or 0.9% of the total in that region for JULES-RothC and decreased by 6.5 Pg C or 0.9% of the total in that region for JULES-Q10. Figure 13 compares these spin-up simulations with the simulated change in soil carbon over the 20[th] century, both globally and for the permafrost region. Also

15 shown are two additional 20[th] century simulations, one made directly after the permafrost carbon was initialised ('20[th] century climate + permafrost carbon') and one where the models were re-spun up for the additional 25 x 20 years after the permafrost carbon was initialised ('20[th] century climate + permafrost carbon + further spin up'). In all cases the change over the 20[th] century is larger than the trend in the spin up. For the global simulations, adding the carbon in the permafrost region has little impact on the total simulated change. The very slightly smaller increase in soil carbon is small compared with the differences

20 between the model configurations. In the permafrost region, the overall increase in soil carbon is smaller, and as expected the addition of permafrost carbon has a greater impact on the changes.

## 4 Conclusions

This paper presents a vertically resolved model of soil carbon developed as a precursor to adding the permafrost carbon feedback into UKESM. This new model includes a tracer so that specified soil carbon (such as permafrost carbon) can be

25 identified, labelled and followed through the simulation. The vertically resolved model improves the spatial representation of soil carbon when compared to the standard non-vertically resolved model. The seasonal peak of the soil respiration in the Northern Hemisphere summer is delayed by leading to the Net Ecosystem Exchange peaking slightly earlier in the year. Once included within an ESM, the change in seasonal cycle will feed back onto the model simulated climate. The change in soil carbon over the 20[th] century is comparable with the standard model.

30 Given the two temperature-dependent rate-modifying functions available in JULES, our results suggest that the RothC temperature function, $F_{T,Roth}$ (Equation 7), should be used for the vertically discretised version, in preference to the $F_{T,Q10}$ (Equation 8), as it gives a better match with the observations both for soil carbon distribution and the seasonal cycle of soil respiration. However, there may be some compensating errors due to the incorrect north-south gradient in NPP. The exact nature





of the most relevant soil temperature $[F_T(T_{soil})]$ and soil moisture $[F_s(s)]$ functions and their applicability across different biomes needs further investigation (Exbrayat et al., 2013).

Two of the most notable remaining sources of model errors are (1) errors in the model simulation of the litter input and (2) missing vertical processes within the soil carbon model such as alluvial sedimentation; dust deposition; peat development.

These should all be considered in future model developments. In order to reduce the influence of errors caused by litter, an alternative method of model evaluation may be to use soil carbon turnover times analogous to those used by Carvalhais et al. (2014). These may not be appropriate for simulations of permafrost carbon, but may be useful in determining the behaviour of the soil carbon within the active layer. Hugelius et al. (2016) suggested the observed carbon stocks could be categorised based on whether the appropriate process is included within an ESM and hence whether the ESM is expected to simulate soil carbon

in that region.

Initialisation of the deep soil carbon according to the NCSCDv2 map allowed a better match between the simulated soil carbon and the observed carbon distribution. Although there is a slight drift in the regional and global soil carbon, this is small compared with the change in soil carbon over the 20[th] century. This methodology may provide a way of initialising an ESM so that the permafrost carbon feedback is more appropriately estimated.

## Code Availability

The JULES code used in these simulations is available from the Met Office Science Repository Service at

https://code.metoffice.gov.uk/trac/jules/browser/main/branches/dev/eleanorburke/vn4.3_permafrost (registration required).

## Competing Interests

The authors declare that they have no conflict of interest.

*Acknowledgements.*  The authors acknowledge the financial support by the European Union FP7-ENVIRONMENT project PAGE21 under Contract GA28270 and the European Union Horizon 2020 Project CRESCENDO under Contract 641816. Eleanor Burke was supported by the Joint UK BEIS/Defra Met Office Hadley Centre Climate Programme (GA01101). Sarah Chadburn was supported by the Joint Partnership Initiative project COnstraining Uncertainties in the Permafrost-climate feedback (COUP) (National Environment Research Council grant NE/M01990X/1). We also acknowledge the support of Dr Gustaf Hugelius who post-processed the WISE30sec data into netcdf form.



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



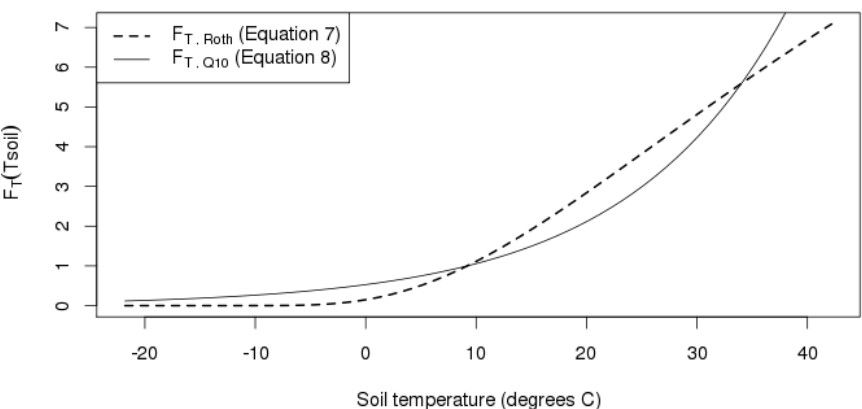

**Figure 1.** The temperature response curves $[F_T(T_{soil})]$ from Equation 7 and Equation 8.

| Region | WISE30sec | NCSCDv2 | **JULES-Roth** | JULES-Roth$_{onelyr}$ | **JULES-Q10** | JULES-Q10$_{onelyr}$ |
|---|---|---|---|---|---|---|
| Global | 1943 | | **2545** | 1259 | **2976** | 1311 |
| Permafrost | 452 | 741 | **568** | 117 | **325** | 90 |
| Tropical | 542 | | **491** | 356 | **832** | 446 |
| Tropical forest | 328 | | **293** | 218 | **493** | 274 |
| Boreal forest | 567 | | **959** | 325 | **759** | 275 |
| Tundra | 182 | 292 | **126** | 37 | **72** | 26 |

**Table 1.** Total soil carbon in top 2 m (Pg C) for the regions assessed in this paper (top three lines) and for the following biomes defined by Olson (2001): tropical forest; boreal and coniferous forest; and tundra (bottom three lines).

Zimov, S. A., Davydov, S. P., Zimova, G. M., Davydova, A. I., Schuur, E. A. G., Dutta, K., and Chapin, F. S.: Permafrost carbon: Stock and decomposability of a globally significant carbon pool, Geophysical Research Letters, 33, n/a–n/a, doi:10.1029/2006GL027484, http://dx.doi.org/10.1029/2006GL027484, l20502, 2006.





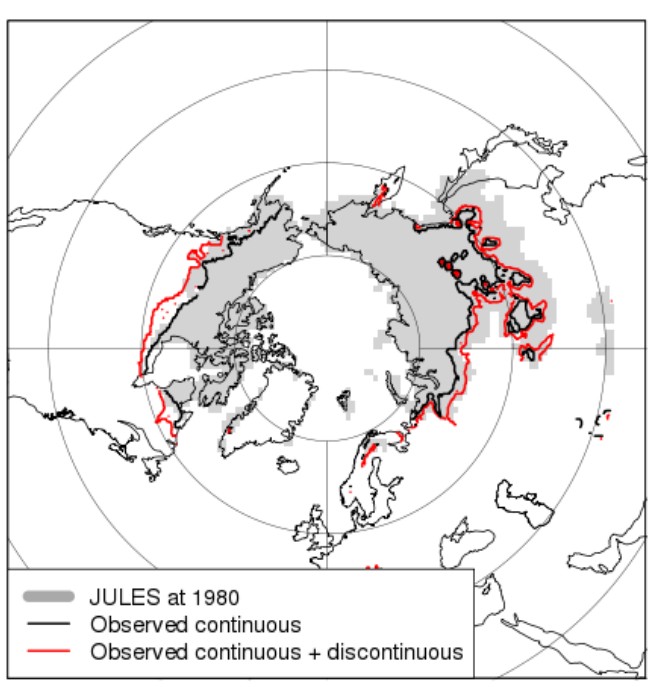

**Figure 2.** JULES simulated permafrost extent is shaded grey. The boundaries of the observed continuous and discontinuous permafrost are superimposed in black and red.

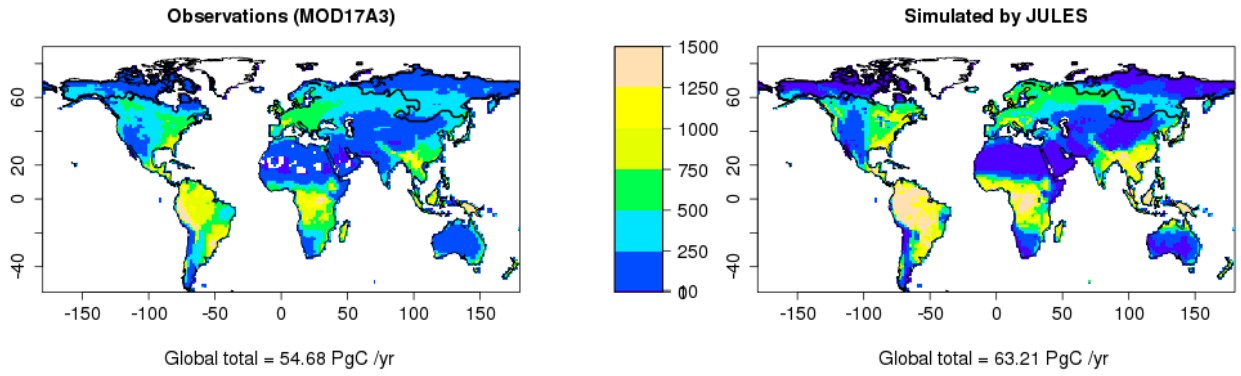

**Figure 3.** MODIS observed and JULES simulated multiannual mean Net Primary Productivity in g C m$^{-2}$ year$^{-1}$ for the period 2000-2012. The black contours highlight the region where JULES simulates permafrost.



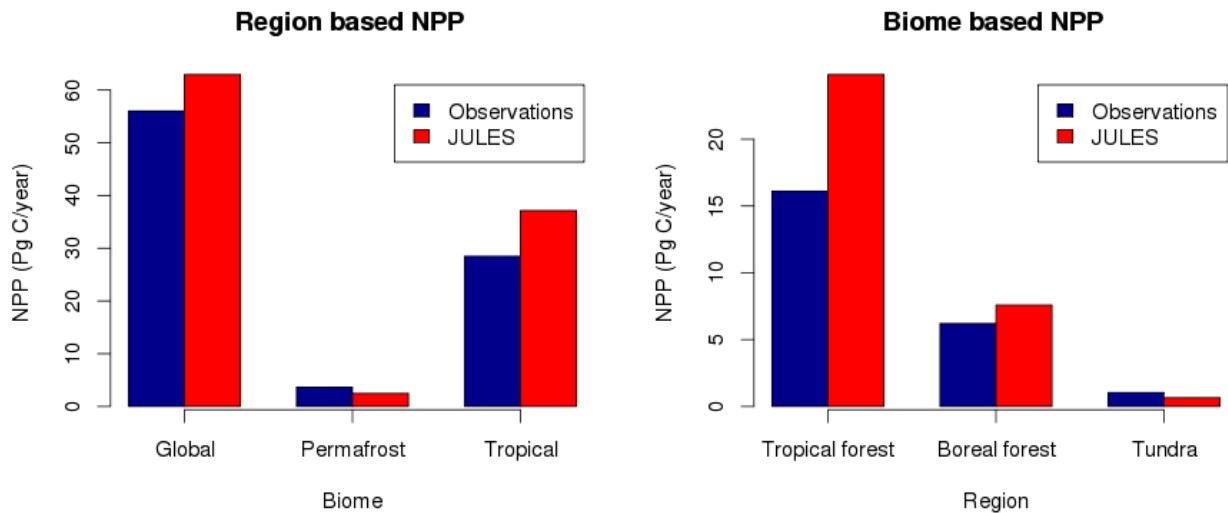

**Figure 4.** MODIS observed and JULES simulated annual mean NPP for the regions assessed in this paper (left hand plot); and for the following observed biomes defined by Olson et al. (2001): tropical forest; boreal and coniferous forest; and tundra (right hand plot). Note the different scales.

| Vertically resolved simulations | NCSCDv2 | JULES-Roth | JULES-Q10 |
|---|---|---|---|
| Mean Ratio: deep / total | 0.59 | 0.53 | 0.60 |
| St. dev. of Ratio: deep / total | 0.11 | 0.07 | 0.06 |
| Total soil carbon (Pg C) | 1007 | 801 | 446 |
| Labelled deep soil carbon (Pg C) | 585 | 475 | 251 |
| **Added obs. of deep carbon** | | JULES-Roth | JULES-Q10 |
| Mean Ratio: deep / total | | 0.76 | 0.81 |
| St. dev. of Ratio: deep / total | | 0.25 | 0.21 |
| Total soil (Pg C) | | 855 | 724 |
| Labelled deep soil carbon (Pg C) | | 543 | 528 |

**Table 2.** Labelled deep soil carbon and total soil carbon in the permafrost region before and after adding the NCSCDv2 observed deep soil carbon for depths below 1 m. Any differences between NCSCDv2 and the added labelled deep soil carbon are caused by differences between interpolation methodologies.





**Figure 5.** Soil carbon in the top 2 m in kg m$^{-2}$ for the 4 different model simulations. The new vertically discretised model versions are on the left and the standard model versions are on the right. The WISE30sec observed data set is shown at the bottom.





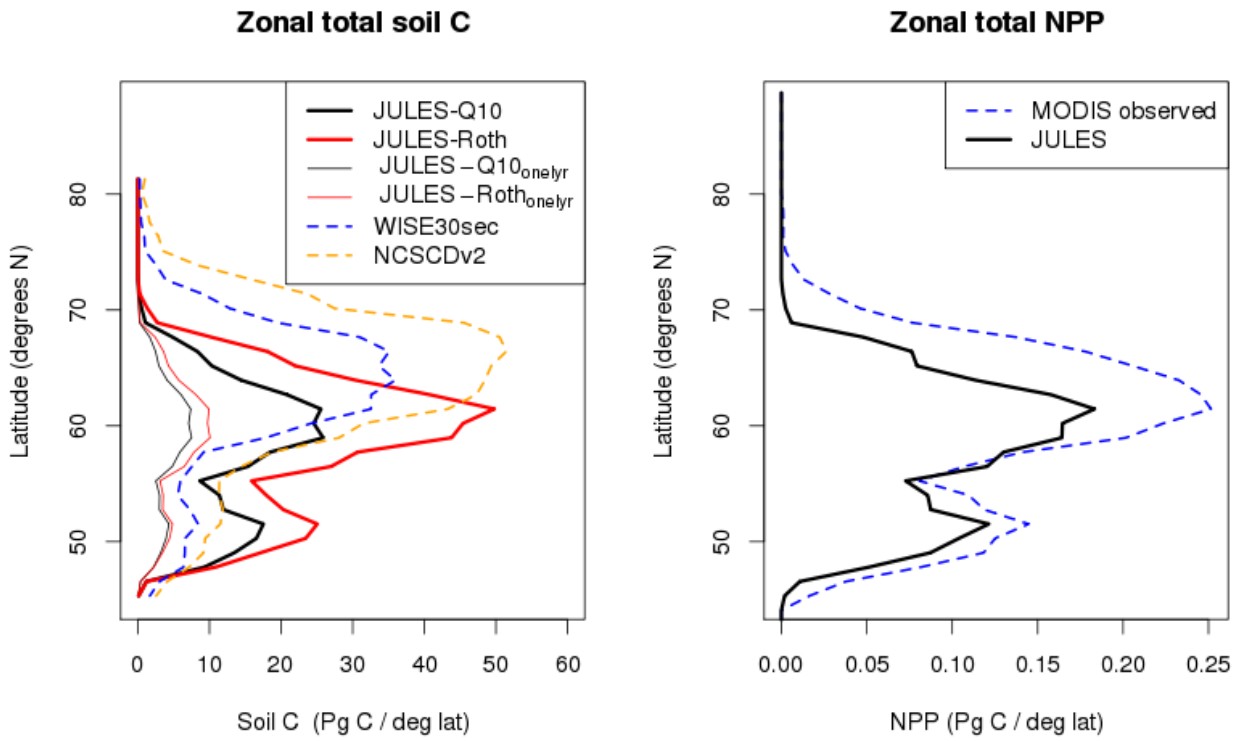

**Figure 6.** Zonal total of soil carbon (left hand plot) and NPP (right hand plot) for the permafrost region highlighted in Figure 3, expressed as Pg C per degree of latitude.





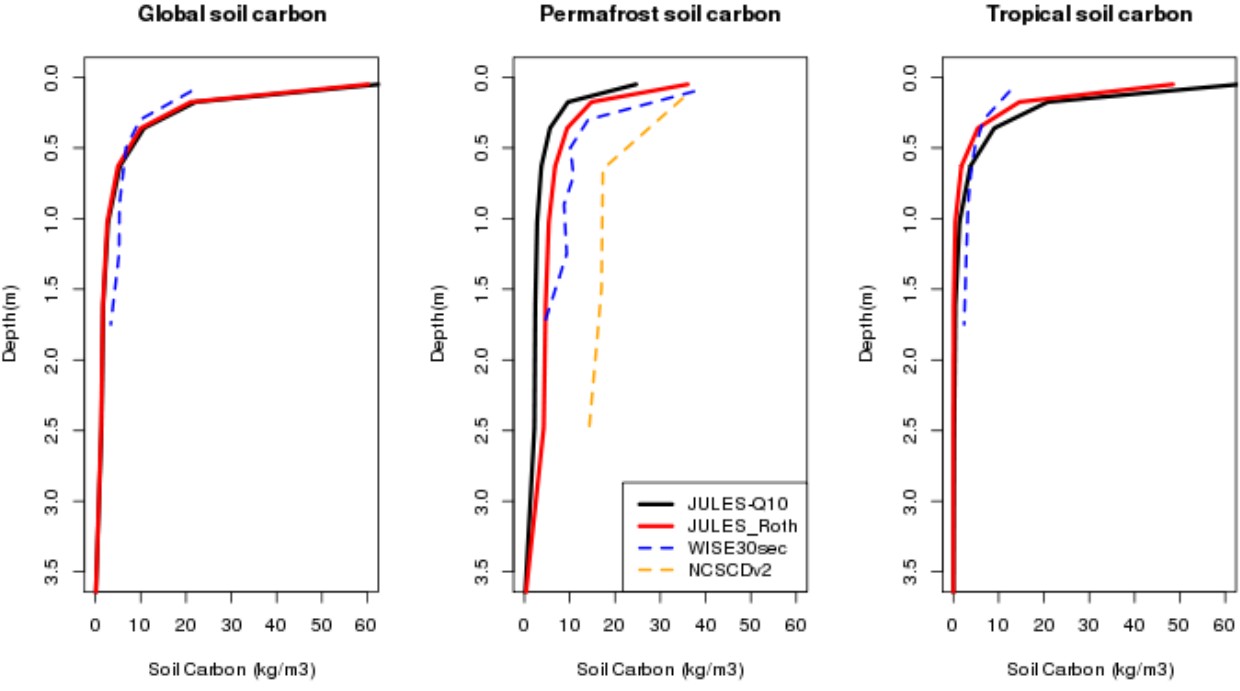

**Figure 7.** Profile of soil carbon in kg m$^{-3}$ for the three main regions in this study.





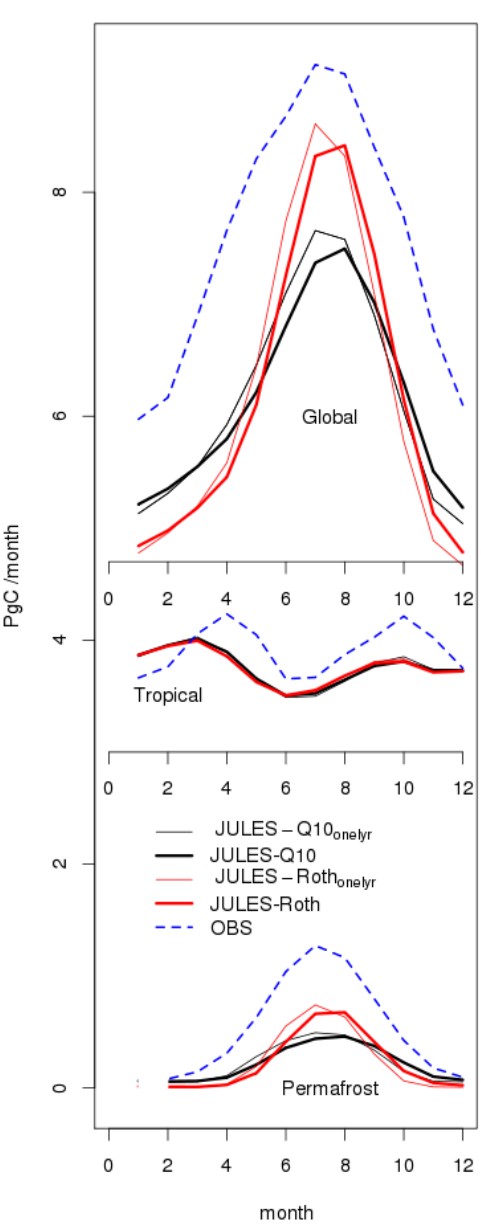

**Figure 8.** Seasonal cycle of total monthly respiration for the three main regions considered in this paper.





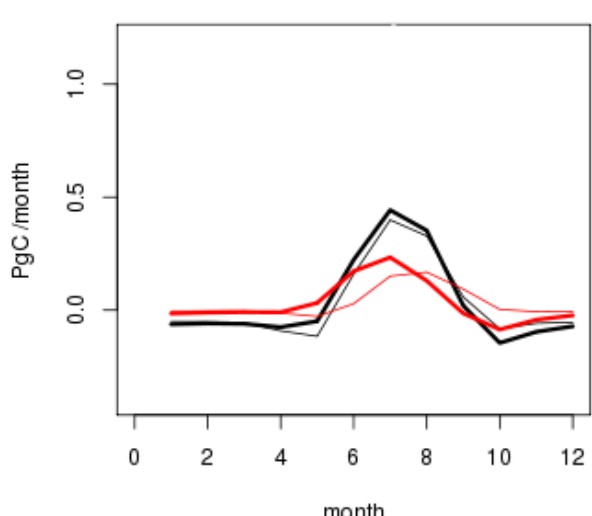

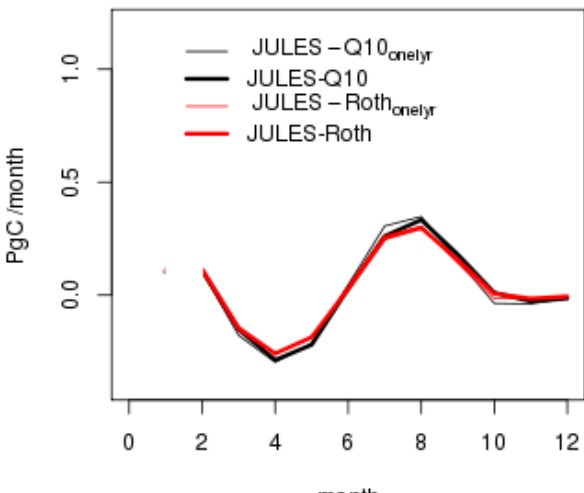

**Figure 9.** Seasonal cycle of Net Ecosystem Exchange (NEE) for the three main regions considered in this paper. Positive values represent an uptake of carbon and negative values represent a loss of carbon.





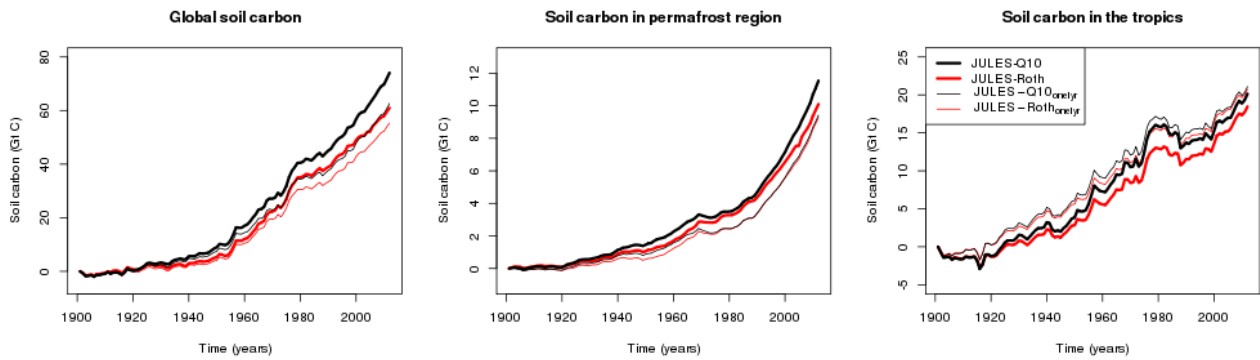

**Figure 10.** Time series of change in soil carbon (in Pg C) over the 20th century for the three regions: global, permafrost and tropical.





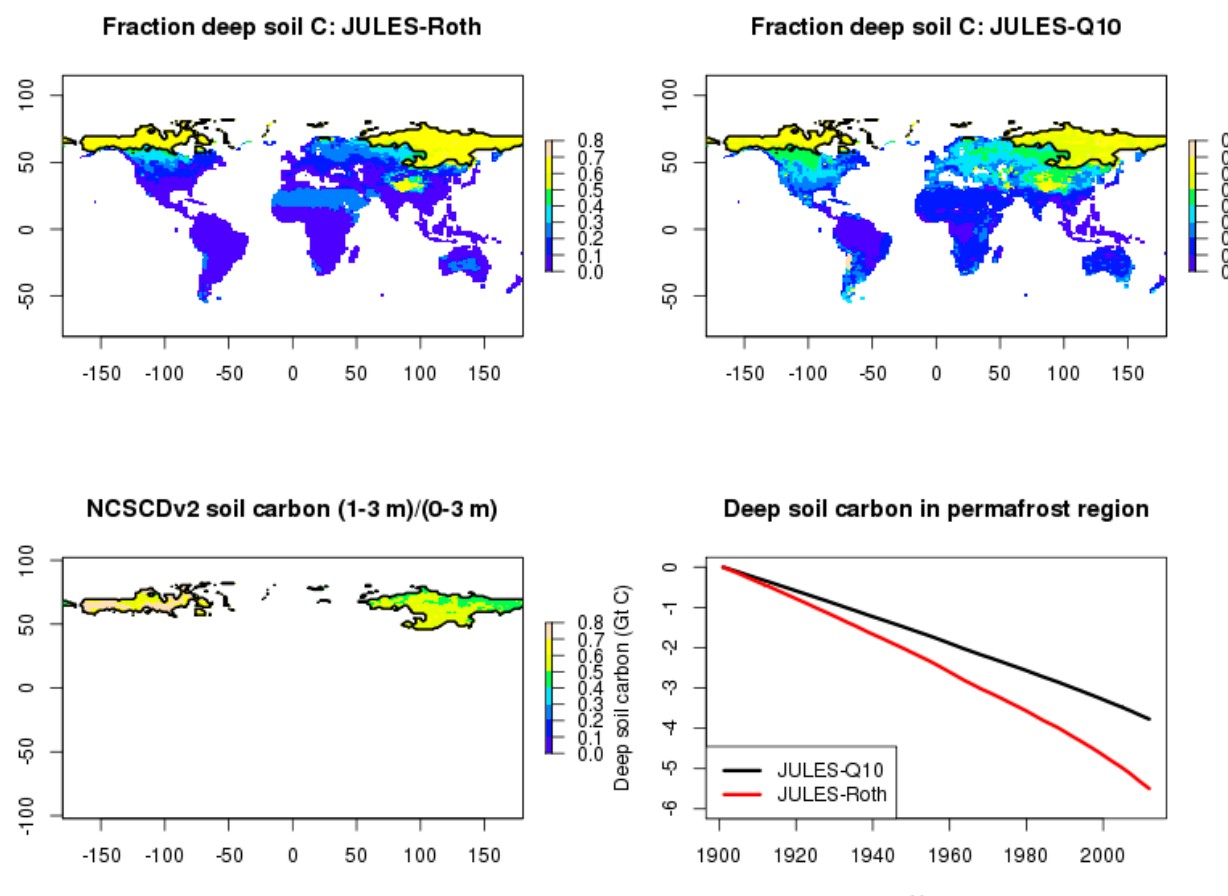

**Figure 11.** The spatial plots show the deep soil carbon (defined as soil carbon below 1 m in 1901) as a fraction of the total soil carbon for each grid cell. The time series shows the change in deep soil carbon for the permafrost region in Pg C.





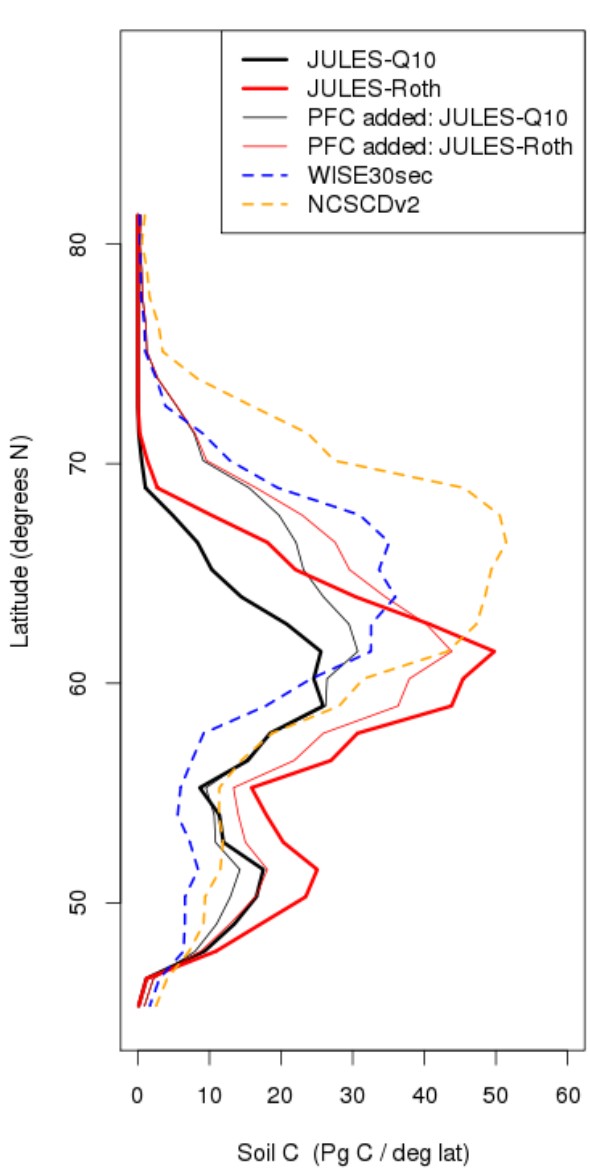

**Figure 12.** Zonal profile of total soil carbon after re-initialisation with observed soil carbon at depths between 1 and 3 m - compared with observations, and with the vertically resolved versions of JULES-Q10 and JULES-Roth.





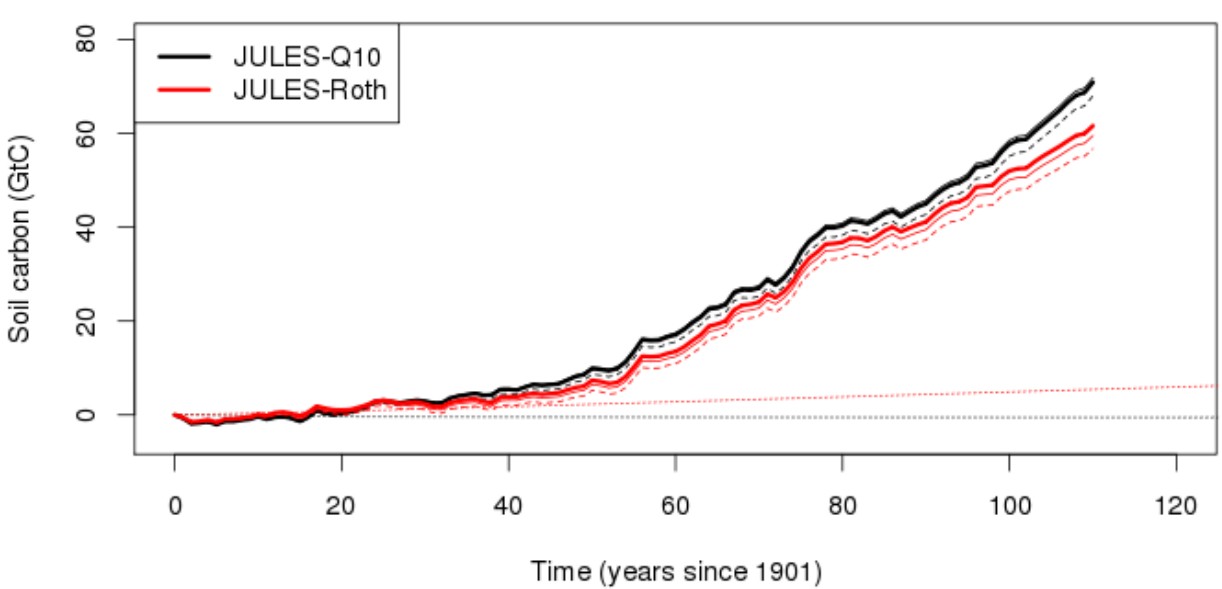

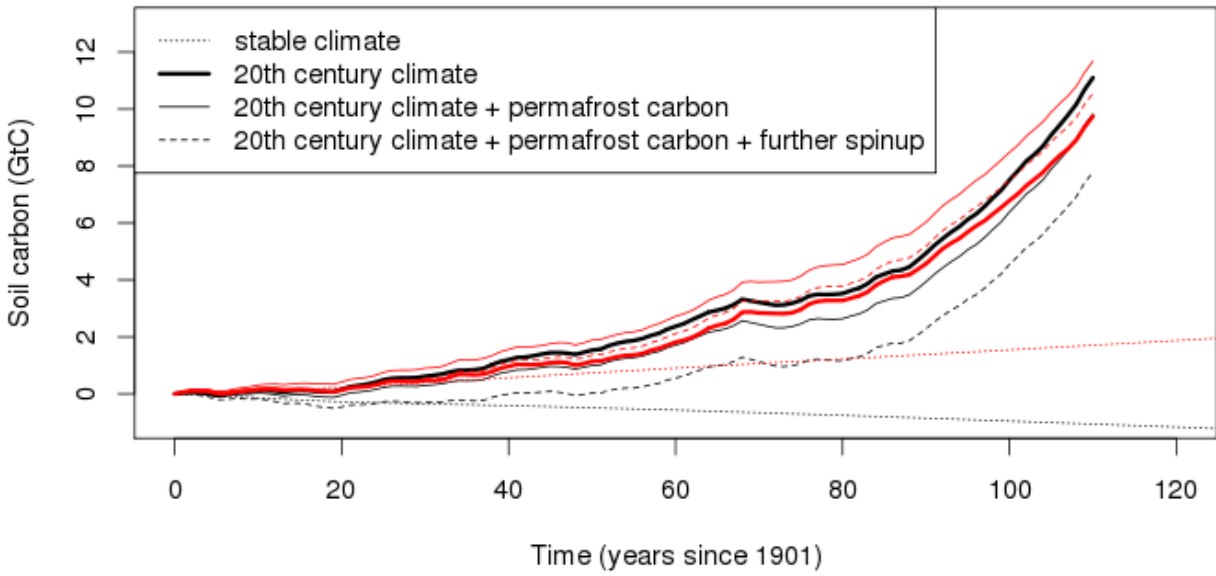

**Figure 13.** Change in soil carbon over 20[th] century with and without the observed permafrost carbon added, and with and without further spin-up.