# Peer review of "A vertical representation of soil carbon in the JULES land surface scheme (vn4.3\_permafrost) with a focus on permafrost regions"

_Geoscientific Model Development, 2016_

## Referee Comment (RC1) · Anonymous Referee #1 · 11 Oct 2016

The paper by Burke et al describes the results of including a vertical dimension to the soil carbon cycling in the JULES model, with particular application to the cycling of carbon in permafrost-affected soils. Including some representation of the vertical carbon dynamics is an essential prerequisite to consideration of permafrost carbon feedbacks and thus represents an important evolution in the terrestrial carbon models used in ESMs. The only real downsides to doing so are (a) the increased complexity of the model, with new uncertainty on things like vertical soil carbon transport and an increased sensitivity to soil climate below the surface, and (b) some more complex logistics with respect to spinup and how to treat the initial conditions which may contain remnant carbon from prior climate states. Since both of these represent real uncer-

tainties in the high latitude system, and since the single layer approach that has been almost universally used until recently leads to a biased set of outcomes to the carbon-climate feedback prediction, formally including these uncertain processes is warranted in the models. This manuscript is a nice description of the methods that have been followed by the JULES group–one of the more widely-used ESMs globally–in making this change to vertically-resolved soil carbon schemes, and I thus recommend its publication.

Some specific comments questions below:

equation 6 and related text: if s is the unfrozen soil moisture, does that mean that, when soils freeze, s drops below s_min and therefore $F_s = 0.2$? If so, then that would contribute a fairly strong indirect temperature sensitivity via the soil moisture mechanism in frozen soils, which would be multiplicative of the direct temperature sensitivity, and so this ought to be noted here. This is how the freeze inhibition works in some other models as well, e.g., CLM, and is a reasonably mechanistic way of including the direct freezing inhibition. Though if that is the case here, I'd suspect that the minimum $F_s = 0.2$ parameter exerts reasonably strong control on the permafrost carbon stocks, at least in the Q10 case where respiration rates are otherwise nonzero in frozen soils, and so you may want to include that in your sensitivity analysis.

Intro to section 2.3: Does the prognostic soil carbon described in this manuscript feed back on the physical parameters from the Chadburn et al organic soils treatment? I am suspecting not, as you state there are no feedbacks from the soil onto the rest of the ecosystem model on line 20 of page 8, but want to confirm that that is the case.

Figure 3 and associated text: Could you change the color bar to give some sensitivity at the lower range of the NPP spectrum as appropriate to the tundra? In the current figure, it is unclear how and where the modeled and MODIS tundra NPPs differ, but this is crucial in understanding the soil carbon predictions from the model.

Figure 4: Whys isn't the NCSCD shown here as it is elsewhere? I'm not understanding

exactly the distinction in usage that is being made between the different soil carbon maps here.

Page 13, line 1: is this partitioning happening per soil level as well as per gridcell?

Figure 11: I am a little confused about what we are supposed to learn from this. If you track soil carbon that is initially in the system, then it will tend to decrease in time under steady conditions as the old carbon leaves the system and is replaced by new carbon. So there is information about the timescale of exchange between the deep carbon and the atmosphere here, but that isn't how you are presenting the data. E.g., if so, then you'd want to separate out the changes due to the transient boundary conditions. So I suggest clarifying what this result means.

The authors point out that an issue with initializing the deep carbon is that the model will tend to drift back to an equilibrium state. I'd point out a second issue, which is that doing so requires a much more careful treatment of how we go about benchmarking the models. Since a dataset cannot logically be used for both initialization and benchmarking, if we initialize the model with observations then we lose one of the few constraints we have on whether the model's soil carbon scheme is reasonable or not. Note that I am not accusing the present manuscript of circularity, as the authors do a nice job of separating out the comparisons of models when uninitialized versus the dynamics of the model when it is initialized later in the manuscript, but I'd recommend being very clear, e.g., in CMIP-type activities if this version is used in them, to specify when the model is initialized using specific soil carbon datasets versus allowed to find its own equilibrium, so that users of the model output do not fall into this kind of circularity trap.

---

## Short Comment (SC1) · 24 Oct 2016

Dear authors,

In my role as Executive editor of GMD, I would like to bring to your attention our Editorial version 1.1:

http://www.geosci-model-dev.net/8/3487/2015/gmd-8-3487-2015.html

This highlights some requirements of papers published in GMD, which is also available on the GMD website in the 'Manuscript Types' section:

http://www.geoscientific-model-development.net/submission/manuscript_types.html

[Figure]

In particular, please note that for your paper, the following requirement has not been met in the Discussions paper:

- "The main paper must give the model name and version number (or other unique identifier) in the title."

Please add a version identifier you use throughout the paper (vn4.3_permafrost) to the title upon your revised submission to GMD.

Yours,

Astrid Kerkweg
* * *

---

## Referee Comment (RC2) · Anonymous Referee #2 · 27 Oct 2016

Review of gmd-2016-235: A vertical representation of soil carbon in the JULES land surface scheme with a focus on permafrost regions

The article presents a clear and well-structured description of a new addition to the JULES land surface model which includes a vertically resolved treatment of soil carbon stocks, transport and reactivity. Results are emphasized for the permafrost regions in comparison to the one-layer versions of the JULES model and existing observational datasets. The addition of vertical resolution in the soil zone generates overall larger estimates of soil carbon and a delay in the onset of respiration resulting from the prop- agation of temperature through the soil profile. These changes generally appear to result in closer agreement with observation, through they are heavily contingent upon

the parameterization necessary for both the ROTH and Q10 versions of the model.

The paper is generally well organized and clear. The description of vertical resolution (e.g. eq. 10 – 14) is critical to the novelty of the work and should be expanded to include a more thorough description of the diffusion coefficient, how it is parameterized and the nature of the depth dependence assigned to functions within eq. 14. In particular the transfer of respired C through the soil profile is not discussed, but this has direct implications for the use of the 'oldC' tracer.

Specific (in line) edits:

P1 L 11: 'not all of the processes relevant for the accumulation of permafrost carbon are included' – give more details for this statement

P4 Eq. 7 and 8 – flip order to agree with text

P4 L25: presumably both temperature functions are evaluated for a given choice of parameters for both soil moisture and vegetation. This assumes that the same relationship between choices of Ft would hold at a different choice of parameters for the other factors Fs and Fv – i.e. the relationship between Ft, Fs and Fv is linear. Is this the case?

P5 eq. 12-13: clarify is depth dependence assigned to Beta_ïĄćR or is the (z) assigned to the overall change in R_tot with depth? Is an explicit treatment of the depth dependent fraction of soil respiration emitted to the atmosphere necessary?

P5 L26: As written in eq. 10 – 13 D(z) is not a 'diffusion rate' it is a diffusion coefficient in units of L2/t

P6 L20: If the fraction of labelled C in the respiration flux is being used, then the earlier issue of how this C is allowed to move through the depth-resolved profile (e.g. Beta_R) needs to be clarified. It would seem that this parameter would become the fraction of respired C from each depth interval that reaches the interval above, and which should be added to respired C in that interval
[Figure]

P7 L5: define PFT

P7 L10: 'are shown here' – refer to a figure number or rephrase

P 7 L23: 'soil carbon pools rescaled by relevant factors' – clarify are these the same factors the diffusion coefficients and rate constants were scaled by?

P9 L15: why 'and vice versa'?

P9 L25: Some comment could be made here to reconcile these results with the earlier statement (P2 L26) that 'Without a vertical representation, decomposition rates are determined only by soil temperatures above the maximum summer thaw depth, so the very slow turnover of deep carbon in the permanently frozen soil is not represented'. This statement would seem to suggest that the vertically resolved model should predict a lower value, in contradiction with what's shown here.

P9 L30: elaborate on this statement 'soil carbon is not in equilibrium with the current climate' – the model was initialized for the period prior to the current climate. . .

P11 L5: This is a nice result, though still would like more clarification on the treatment of respired C diffusion.

P12 L5: more discussion about implications for imposed difference in ïĄťresp between models on results would be helpful

---

## Author Comment (AC1) · 14 Dec 2016

A vertical representation of soil carbon in the JULES (vn4.3_permafrost) land surface scheme with a focus on the permafrost regions.

The title has been changed to the above as requested by the editor.

The authors thank the reviewers for the clear and interesting reviews. Their comments will be addressed in the following manner.

Referee 1

Equation 6 and related text: Indeed the soil moisture used in Equation 6 reduces when

[Figure]

Interactive
comment

the soil freezes and hence provides an additional constraint on the temperature sensitivity. As with many other models, JULES has a proportion of unfrozen water even at temperatures below freezing (Cox et al., 1999, Figure 3a). This is dependent on the soil type as is the wilting point, so the minimum soil moisture threshold will vary with location as well as time of year. This paper has only tested a couple of the many available parameterisations of the control of temperature and soil moisture on respiration. It would be very informative to carry out a systematically sensitivity study of these parameterisations in order to get a more definitive suggestion as to which are most suitable. This will be the subject of further work. I have changed the order of Equations 6 to 8 and added information on the freeze inhibition as a function of temperature and the additional constraint on the temperature sensitivity by the moisture function.

Intro to section 2.3: The prognostic soil carbon does not feedback onto the organic soil characteristics. This is in the queue of further developments. This lack of feedback has been made clearer on line 20 of page 8.

Figure 3 and associated text: We agree, the NPP of the northern high latitudes is not clear in this figure. The colour bar has been changed to clarify this.

Figure 5: We have added the NCSCD data as a new subplot to Figure 4. (It wasn't there because in an earlier draft we showed the soil carbon scaled by the global mean and couldn't do an equivalent thing with the NCSCD). The caption has been expanded to clarify the Figure and the discussion at the top of page 10 expanded. Page 13, line 1: Yes the partitioning of the soil carbon between pools was done by interpolating the observations to the model depths and partitioning per soil level. This is clarified in the revised document.

Figure 11: In the permafrost the deep soil carbon has a very slow turnover times. This is slow enough for it to be assumed to be inert on timescale of 100-1000 years. Therefore in an equilibrium simulation there is little exchange of soil carbon between the deep soil and the shallower soil. However, the climate and $CO_2$ changes during a 20th

century transient run mean that there is some transfer between the deep and shallow soil carbon as is described here. This is a simple representation of 'old' permafrost carbon which is traced when permafrost thaws in a changing climate and is used in Burke et al., (2017. to be submitted to Bogeosciences) to identify the soil carbon related to the permafrost feedback. There is also great potential for a model tracer such as the one discussed here to be used for model evaluation using radioisotopes (He, Y., Trumbore, S.E., Torn, M.S., Harden, J.W., Vaughn, L.J., Allison, S.D. and Randerson, J.T., 2016. Radiocarbon constraints imply reduced carbon uptake by soils during the 21st century. Science, 353(6306), pp.1419-1424.). The discussion and caption around Figure 11 have been modified to clarify the fact that the time series of deep soil carbon shown only includes the original deep carbon, not the increase in the soil carbon at depth, analogous to the loss of permafrost carbon.

Initialising soil carbon: A discussion of the issue of benchmarking models given that they have already used the observed data set for initialisation is added to the text. This paper mainly evaluates the model using the observed soil carbon. However, we feel that there is a better method of constraining models based on metrics such as turnover times. This is the subject of on going work.

Referee 2

The description of the vertical transfers in the soil carbon model is expanded and clarified. In particular equations for the soil carbon mixing are added and the depth dependence for each function in Equation 14 more clearly defined. FT and Fs are dependent on the soil temperature and moisture profile calculated within JULES.

The transport of the respired carbon through the system is not yet included in this model version. Once respired the carbon is assumed to be instantly transferred to the atmosphere. Given that the soil carbon is assumed to be emitted as CO2 rather than CH4, its form is unlikely to change in the soil column, so the amount will be roughly correct. However, there will be some delay from time of decomposition to

time of emission from the soil surface. This delay will increase with increasing depth. In addition some gas may be temporarily or more permanently held within the soil structure. We are uncertain as to how much impact adding this process will have on the model results and this is another development in the pipeline for the future. This will become more important when the respired carbon is differentiated between $CO_2$ and $CH_4$. $CH_4$ is readily oxidised as it is transported through the soil profile. This is another required model development.

Any old carbon respired will also be instantly released to the atmosphere. The old soil carbon moves throughout the profile in a similar way to the rest of the soil carbon i.e. via mixing. This is clarified in the document.

P1, L11: more details added on the missing processes such as peat accumulation.

P4, Eq 7 & 8: order flipped

P4, L25: Only a limited evaluation of the different functions limiting respiration is discussed here. As discussed in "Equation 6 and related text" above the soil moisture function also includes some indirect temperature sensitivity. Although beyond the scope of this current paper, a fuller analysis using a large model ensemble with a range of FT and Fs would enable a range of simulated soil carbon distributions to be assessed more carefully. I think one of the conclusions which should be added to this paper (see also "Initialising soil carbon") is that alternative methods of constraining the present day soil carbon distribution need to be developed in order to more rigorously assess model output.

P5, Eq 12, 13: Beta_R has no depth dependence. Rtot has depth dependence because it is the sum of the respiration from all of the pools, i.e. the sum of the Ri in Equation 14. There is no explicit treatment of transfer of the respired carbon to the atmosphere – this is discussed above in more detail.

P5, L26: diffusion rate is changed to diffusion coefficient.

P6, L20: The 'old' carbon transfers through the profile in the same manner as the rest of the soil carbon. The respired 'old' carbon is instantly transferred tot he atmosphere.

P7, L5: PFT – Plant Functional Type – is defined.

P7, L10: This sentence is reworded to "Two different model simulations were carried out using two alternative parameterisations of soil respiration."

P7, L23: These are the same factors the diffusion coefficients and rate constant were scaled by. This is added to the text.

P9, L15: too high NPP means too much input of organic carbon to the soil resulting in a high soil carbon.

P9, L25: The equilibrium state of soil carbon depends on a balance between the inputs and the decomposition rate. The model take a long time to reach this equilibrium because the decomposition rates and the diffusion rates of the 4th pool are very slow (hence the need to accelerate the soil carbon spinup), but a significant amount of soil carbon is built up in the deeper soils. A sentence elaborating the difference between spin up and 20th century climate is added.

P9, L30: JULES generates its soil carbon stocks by running for many years until the soil carbon is in steady state. The input (NPP) is balanced by the output (respiration). The observations contain this equilibrium soil carbon plus further non-equilibrium soil carbon such as is in peatland and waterlogged soils. JULES is not expected to be able tor represent this non-equilibrium soil carbon (Hugelius et al., in preparation). The last sentence on page 9 has been updated to more carefully reflect this discussion.

P11, L5: As discussed above, we have not yet included the transport of respired carbon to the atmosphere but plan on including this and exploring its impact in the future.

P12, L5: There is an interesting interrelationship between tau_resp and FT and how they relate to the modelled soil carbon. This is discussed in more detail in this section and in the discussion of Figure 5. tau_resp exerts strong control over the amount of

soil carbon in the profile. It does not impact the difference between the tropics and the northern high latitudes. In contrast FT affects the relative amounts of soil carbon in the cold and the warm regions.